# Histopathological Study on Collagen in Full-Thickness Wound Healing in Fraser’s Dolphins (*Lagenodelphis hosei*)

**DOI:** 10.3390/ani13101681

**Published:** 2023-05-18

**Authors:** Chen-Yi Su, Tzu-Yu Liu, Hao-Ven Wang, Wei-Cheng Yang

**Affiliations:** 1School of Veterinary Medicine, National Taiwan University, Taipei 106216, Taiwan; angelsu2096@gmail.com; 2Department of Life Sciences, National Cheng Kung University, Tainan 701, Taiwan; brothansis@yahoo.com.tw; 3Marine Biology and Cetacean Research Center, National Cheng Kung University, Tainan 701, Taiwan; 4Center for Bioscience and Biotechnology, National Cheng Kung University, Tainan 701, Taiwan

**Keywords:** type I collagen, type III collagen, wound healing, Fraser’s dolphins

## Abstract

**Simple Summary:**

The deposition of excessive collagen resulting in scarring is a detrimental outcome of the wound healing process in humans, ultimately impeding the successful restoration of skin function. Surprisingly, in Fraser’s dolphins (*Lagenodelphis hosei*), collagens undergo a configuration change during wound healing and eventually recover to normal architecture. However, little research has been conducted on dolphins’ collagen composition in the skin and the underlying mechanism during the wound healing process. Here, histochemical staining and immunostaining were performed on normal and wounded skin samples in different healing stages of Fraser’s dolphins. The results demonstrate that collagens changed in composition during the wound-healing process. The mature healed wound implies the great ability to remove excessive collagen depositions in hypertrophic-like scars in Fraser’s dolphins. A better understanding of collagen dynamics in Fraser’s dolphins would provide more information and possibilities for solving aberrant scarring problems in humans.

**Abstract:**

Fraser’s dolphins (*Lagenodelphis hosei*) possess great healing abilities. Their skin composition can be restored after wounding, including collagen spacing, orientation, and bundle thickness. However, it remains unclear how collagens are involved in the wound-healing process and eventually regain normality in Fraser’s dolphins. Learned from the other two scarless healing animals, changes in type III/I collagen composition are believed to modulate the wound healing process and influence the scarring or scarless fate determination in human fetal skin and spiny mouse skin. In the current study, Herovici’s, trichrome, and immunofluorescence staining were used on normal and wounded skin samples in Fraser’s dolphins. The results suggested that type I collagens were the main type of collagens in the normal skin of Fraser’s dolphins, while type III collagens were barely seen. During the wound healing process, type III collagens showed at early wound healing stages, and type I collagen increased in the mature healed wound. In an early healed wound, collagens were organized in a parallel manner, showing a transient hypertrophic-like scar, and eventually restored to normal collagen configuration and adipocyte distribution in the mature healed wound. The remarkable ability to remove excessive collagens merits further investigation to provide new insights into clinical wound management.

## 1. Introduction

Extracellular matrix (ECM) is an essential component of skin, interlocking different molecules for structural scaffolding, maintaining tissue homeostasis, and modulating wound healing [1,2]. ECM is the core of matrisome, the ensemble of genes encoding ECM and ECM-associated proteins, forming a three-dimensional network of macromolecules with various proteins. For interstitial connective tissue, ECMs, such as collagen, gelatin, laminin, and fibronectin, provide structural scaffolding for tissue and organize orientation for cells with flexibility and stretching ability. In the basement membrane, ECMs can bind integrins to regulate multiple biological processes, control morphogenesis, and modulate cellular metabolism. Among ECMs, collagens are the most abundant proteins and are conserved in all multicellular animal kingdoms. The fibril-forming collagen matrix increases with age and contributes to different dermis regions; larger collagen fibrils densely accumulate in the deep reticular region, and finer collagens locate in papillary regions beneath the epidermis. Type I collagen and type III collagen, which are the two major collagen types in the skin, constitute about 80–85% and 15–20% of the total collagens in human skin, respectively [3,4,5,6,7]. Both type I and type III collagen are responsible for maintaining skin structure and tissue integrity, while type III collagen can also provide tensility, flexibility, and softness [2]. Type I collagens form thick and stiff fibrils [8], and type III collagens are crucial for type I collagen fibrillogenesis and may influence the fibril diameter of type I collagen [9,10].

As a key component in ECM, collagen plays a critical role in different wound healing phases. Wound healing is a complex series of reactions achieved through three continuous and overlapping phases: inflammation, proliferation, and remodeling. First, released cytokines stimulate the chemotaxis of neutrophils, leading to the following inflammation. In the inflammation phase, degraded collagen fragments recruit immune cells for the removal of microbes and tissue devitalization. In the proliferative phase, fibroblasts migrate into the wound and produce a stronger, organized matrix of collagens and proteoglycans to gradually replace the loose and highly hydrated provisional matrix, such as fibrin, fibronectin, and hyaluronan [2,11]. Changes in collagen composition are mainly denoted by a considerable shift in the type I to type III collagen ratio, which plays a critical role in fibrillogenesis and collagen bundle construction [12]. During the early phase of wound healing, type III collagens increase slowly and randomly and then be replaced by collagen I later in the proliferative phase and following the remodeling phase [12,13]. In addition, collagen fragments act as angiogenic signals to promote blood vessel development as well [14]. Next, in the remodeling phase with re-epithelization, repaired tissue from ‘typical scar’ regains 50–80% integrity and tensile strength of normal skin in humans. In comparison with unwounded skin, the ‘typical scar’ may be functionally deficient due to collagen differences in fiber size, fibril density, and orientation [2,14]. Abnormal collagen levels and configuration may lead to aberrant scar formation during wound remodeling. Collagen fiber orientation in scars (hypertrophic and keloid scar) is parallel to the epidermal surface, unlike that of normal skin, where the fibers form a three-dimensional basketweave-like network [12,14].

As highly conserved structural proteins, collagens are regarded as the essential components to blubber in cetacean skin as well [15]. In bottlenose dolphins, it has been reported that collagen fibers are present in the dermis, hypodermis (blubber), and subdermal connective tissue sheath from a decomposing individual by polarized light microscopy [16]. The collagen fiber was identified by its compacted structure and three-dimensional configuration, which show the collapse of latticework under chromatographic analysis. In bottlenose dolphins, collagen fibers in the reticular dermis are usually arranged at various angles. As going deeper under the reticular dermis, the number of adipocytes increases gradually, eventually changing into fat tissues (blubber) with few collagen fibers ramifying throughout the blubber. [17]. In addition to the common integument, the tail flukes of the bottlenose dolphin [18] and dorsal keel of the harbor porpoise [19] demonstrate strong and closely interwoven collagen bundles with various angles to the epidermis. The differences in collagen fibers distribution and orientation might contribute to the regional-specific mechanical properties, which were demonstrated in cetacean dorsal keel skin [19]. Additionally, collagen fibers exist in subdermal connective tissue sheath, showing the thickest bundles (30–55 μm) among mammals and interacting with underlying muscle and forming a force transmission system, which might control swimming behavior [17,18,20]. These results indicate the importance of collagens in providing mechanical properties and influencing cetaceans’ physical behaviors.

In addition to regional-specific features in cetacean collagens, temporal-related collagen dynamics are found during the wound-healing process as well. In Fraser’s dolphins, it has been reported that changes in collagen bundle thickness, spacing, and orientation occurred during wound healing [21]. The authors mentioned that in the immature healed wound, collagen fibers are arranged parallel to the surface epidermis, mixing with numerous fibroblasts, blood vessels in the reticular dermis, and blubber. Surprisingly, the full-thickness wounds are completely healed and mostly recovered to normal skin architecture and function, including collagen bundles in thickness, spacing, and orientation, and restoring adipocytes as well. These results indicate the importance of collagen composition in cetacean skin regeneration [21]. The role of collagen dynamics in skin regeneration has been investigated in two scarless healing animals, spiny mice and human fetuses. The spiny mouse, which has been reported to express a higher level of collagen III in the skin than laboratory mice [22], exhibits a remarkable ability to undergo scarless regeneration [23,24]. Type III collagen is the dominant collagen type expressed in the early stages of wound healing in spiny mice and gradually changed later with collagen I highly expressed. A high level of type III collagen in spiny mouse skin is responsible for constituting the soft wound bed for hair neogenesis. In human skin, type I collagen is the major component in both adult and fetal skin. During the wound healing process, type III collagens increase in the early stages of wound healing, and then type I collagen increases later in the remodeling phase in human skin. It is noteworthy that the collagen type III/I ratio in fetuses is higher than in adults [25,26], both in normal skin and recovered skin, and type III collagen remains the main subtype after wounding. The high level of collagen type III may influence the ECM organization and tissue stress. Downregulation of type III collagens leads to increased scar formation. These studies suggest that the type III/I collagen ratio both in unwounded skin (e.g., human fetus vs. adult) and during wound healing (e.g., spiny mouse vs. laboratory mouse) plays an important role in scarring or scarless fate determination. Therefore, we examined the collagen distribution and composition in the normal skin and type I and type III collagen dynamics during wound healing in Fraser’s dolphins. Investigating the collagen alteration process during the wound healing process may help us gain more knowledge of the skin regeneration abilities and pave the way for research on mechanical properties and potential adaptation strategies in response to the aquatic environment in Fraser’s dolphins.

## 2. Materials and Methods

### 2.1. Specimen Collection

Skin samples were collected from six dead-stranded Fraser’s dolphins in Taiwan from 2018 to 2020: two adult males, three sub-adult males, and one calf female. All animal procedures were conducted with the approval of the Ocean Conservation Administration (OAC), Taiwan (Permit #1090002352). The individual information and the number of normal and wounded skin samples used in the current study are shown in Table 1.

### 2.2. Histological Process

Skin samples were fixed with 4% paraformaldehyde (PFA) for trichrome staining and 10% neutral formalin for Herovici’s staining and IHC staining, then dehydrated in graded ethanol and embedded in parafilm. The tissue blocks were cut into 7 μm sections for trichrome and immunohistochemical staining and 5 μm sections for Herovici’s staining.

### 2.3. Trichrome Staining

Sections in 7 μm were incubated in a 60 °C oven for 15 min. After cooling, tissue sections were deparaffinized in xylene and rehydrated in graded ethanol. A commercial kit (Sigma-Aldrich, Burlington, MA, USA), Masson’s trichrome stains was applied. The slides were mordanted in Bouin’s solution at room temperature overnight. The next day, the slides were washed in running tap water to remove the yellow color. Then Weigert’s Iron Hematoxylin working solution was applied for five minutes following running tap water wash. Biebrich Scarlet-acid fuchsin was continually applied for 5 min and washed background over by Phosphotungstic/Phosphomolybdic acid working solution for ten minutes. Then slides were placed in Aniline solution for five minutes with the following 1% acetic acid for two minutes. In the last, slides were rinsed and dehydrated through graded ethanol, cleared in xylene, and mounted in permanent mounting media.

### 2.4. Herovici’s Staining

A commercial kit (American MasterTech, KTHER, Lodi, CA, USA) was applied. Type III collagen and reticular fiber stained blue, type I collagen stained red, cytoplasm stained yellow, and nuclei stained black. Sections in 5 μm were deparaffinized in xylene and rehydrated in graded ethanol first; then, slides were stained with Weigert’s hematoxylin for five minutes and washed in running water for 45 s. Slides were stained with Herovici’s working solution for two minutes and immersed in 1% acetic acid for two minutes. Slides were dehydrated in absolute alcohol three times for one minute each, then cleared in xylene for one minute three times, and mounted with permanent mounting media.

### 2.5. Immunohistochemistry (IHC) Staining

Sections in 7 μm were incubated in a 60 °C oven for one hour. After cooling, tissue sections were deparaffinized in xylene and rehydrated in graded ethanol. Antigen retrieval was performed in Uni-trieve (Innovex Biosciences, Richmond, CA, USA) at 40 °C overnight. The sections were incubated in 3% peroxide at room temperature for 10 min and subsequently blocked with Power Block^TM^ universal blocking reagent (BioGenex Laboratories Inc., Fremont, CA, USA) at room temperature for 10 min. Primary antibodies of type I collagen (ab34710, rabbit polyclonal, Abcam, Cambridge, UK) and type III collagen (NB600-594, rabbit polyclonal, Novus Biologicals, Littleton, CO, USA) were applied in 1:200. Tissue sections were then incubated with primary antibodies overnight at 4 °C. Negative control slides were incubated with PBS only. Liver tissue with fibrotic lesions from a Fraser’s dolphin was used as a positive control. After primary application, a Super sensitive^TM^ polymer-HRP detection system (BioGenex Laboratories Inc.) was used. Super Enhancer^TM^ reagent was applied first at room temperature for 20 min, and then polymer-HRP reagent was continually applied at room temperature for minutes. Phosphate-buffered saline (PBS) was used for rinsing between each step. Tissue sections were then incubated with 3.3′-diaminobenzidine (DAB) substrate solution at room temperature for 10 min, counterstained with Mayer’s hematoxylin for 20 s, and mounted with permanent mounting media.

## 3. Results

### 3.1. Normal Skin

#### 3.1.1. Collagens Distribution in Normal Skin

To investigate the collagen dynamics in Fraser’s dolphin skin, trichrome staining was used to obtain a comprehensive overview of skin composition. In normal skin, collagen bundles are the main components in the upper dermis layer, including the papillary dermis (PD) and reticular dermis (RD); while the hypodermis layer, blubber (B), is mainly composed of adipocytes (Figure 1). Collagen fibrils in blue color in trichrome staining existed in the papillary dermis interdigitating with epidermal rete ridges, and in the reticular dermis underlying the epidermis. The thickness of the reticular dermis layer varies in different body regions and species [17]. In Fraser’s dolphin, reticular dermis thickness (0.2–0.45 mm) in sub-adult dorsal skin (Figure 1) was thinner than in common dolphin (*Delphinus delphis*) (0.7–1 mm) in the back region [17], containing compact and dense collagen fibers with blood vessels interspersed. In blubber, the density of collagen decreased apparently and was replaced with abundant adipocytes.

#### 3.1.2. Type I and Type III Collagen Distribution in Normal Skin

Collagens can be observed as major components of ECM in the dermis by Herovici’s staining as well. The red color here represents type I collagen, while type III collagen is stained blue. In normal skin, type I collagen was uniformly distributed in the reticular dermis and papillary dermis in red color. To further validate the collagen composition, type I, and III collagen were studied by IHC staining. Similar to Herovici’s staining, type I collagen signals existed in the papillary dermis and reticular dermis, while type III collagens were barely seen (Figure 2; Table 2).

In the current study, liver fibrotic tissue from the same dolphin was used as a positive control for type III collagens due to its known deposits in fibrotic tissues such as lung, liver, kidney, and vascular system [27,28].

### 3.2. Wounded Skin

#### 3.2.1. Collagen Distribution during Wound Healing in Trichrome Staining

According to our previous study [21], five stages were classified in wound healing progress in Fraser’s dolphins: Stage 1, now wound; Stage 2, initially healing wound without granulation tissue; Stage 3, healing wound with granulation tissue; Stage 4, immature healed wound with cellular and vascular blubber; Stage 5, mature healed wound with less cellular and vascular blubber. An open healing wound with minimal to moderate granulation tissue is referred to as early stage 3, while a healing wound that is nearly closed is referred to as late stage 3. In early stage 3, thin and dense collagens formed in granulation tissue at the wound edge (Figure 3a,c) in contrast to unwounded skin with abundant adipocytes. At the adjacent area of wounds (Figure 3b,c), collagens in the reticular dermis showed a transition pattern between unwounded skin and wound edge (Figure 3a). In late stage 3, collagens in the wound center showed similar dense and thin collagens as the granulation tissue in the wound edge of early stage 3 (Figure 3c,f). In comparison, collagens in the adjacent area formed a thicker bundle than the center, which indicated more mature stages during wound healing. In stage 4, collagens in an immature healed wound formed organized, thick parallel bundles in the dermis of the wound center, similar to the collagen in the adjacent area of wounds in late stage 3 (Figure 3e,i). These results suggested that newly forming collagen fibrils in granulation tissue (early stage 3: wound edges; late stage 3: wound center) were different from more mature collagens (late stage 3: wound adjacent area; stage 4: wound center) in architecture and configuration. The wound sample of the late stage 3 (Figure 3d–f) was from a calf Fraser’s dolphin, which was different from the other two sub-adult dolphins (Figure 3a–c,g–i). In trichrome staining, different collagen/adipocytes distribution and amounts were observed. The staining results showed a gradual replacement of collagens by adipocytes from the reticular dermis to deeper blubber in a younger individual (Figure 3d). The more mature sub-adult Fraser’s dolphins exhibited a clear boundary between the reticular dermis and blubber (Figure 3a,g).

#### 3.2.2. Type I and Type III Collagen Composition Changes during Wound Healing

To further investigate type I and type III collagen changes during wound healing in Fraser’s dolphin skin, Herovici’s and IHC staining were utilized to examine collagen type (Figure 4). In the stage 3 wound, the upper and middle layer of granulation tissue showed purple fibers mixed with a few blue fibers in Herovici’s staining (Figure 4a). The bottom layer of granulation tissue showed more purple to fuchsia fibers and fewer blue fibers in Herovici’s staining, indicating the increase of type I collagen (Figure 4b). At the adjacent area of wounds, collagens in the reticular dermis were stained fuchsia to red, resembling those of normal skin (Figure 2 and Figure 4c). In IHC staining, stronger type I collagen signals were observed than signals from type III collagens in all three different regions. The increased amount of type I collagen in the adjacent area was observed in IHC staining, while the signal intensity of type III collagen showed no obvious difference in different regions. Hence, the collagen fibers-stained fuchsia to red may indicate an increase in type I collagen in that area (Figure 4c) (Table 3). In addition to the reticular dermis and granulation tissue, purple collagen fibrils existed in the papillary dermis, which interdigitated into the neo-epidermis as well.

In immature healed wounds (stage 4), thin collagen fibers were stained purple to fuchsia and ran parallel in the reticular dermis, resulting from mixed type I and III collagen (Figure 5a). In mature healed wounds (stage 5), collagen fibers were stained fuchsia to red with lower density (Figure 5b). Few thicker collagen fibers also showed in this stage, as in unwounded skin (Figure 5b,c). In addition, the number of adipocytes was higher in the mature healed wound and unwounded skin than in the healing wound at stage 3 or 4 (Table 4).

## 4. Discussion

Previous studies suggest that collagen type III/I ratio plays an important role in wound healing [22,24,25,26] (Figure 6a). In this study, the results demonstrated that type I collagen fibrils were the major components in ECM underlying epidermis in normal skin of Fraser’s dolphin, mainly existing in the papillary dermis and reticular dermis demonstrated by trichrome, Herovici’s and IHC staining. Collagen dynamic with changes in type III/I ratio plays an essential role in ECM remodeling during wound healing, forming thin, disorganized fiber in early wound stages, changing to hypertrophic-like wound with parallel-orientation collagens in the immature healed wound, and eventually restoring to the normal configuration in the mature healed wound (Figure 6b).

Herovici’s staining can differentiate type I and type III collagen [29,30], which has been applied in many different mammals, such as humans [31], rats [32], mice [31,33], and buffaloes [34]. However, it is important to verify the specificity of Herovici’s staining on cetacean tissue samples due to no precedent. Therefore, a comparative analysis of Herovici’s and IHC staining was performed in this study. In Herovici’s staining of the stage 3 wound, the upper and middle layer of granulation tissues contained blue fibers and a few purple fibers, indicating the existence of both type I and type III collagens showed in IHC staining as well (Figure 4a). In comparison, the bottom layer of granulation tissue showed more purple/fuchsia fibers, corresponding with IHC results with more type I collagen signals (Figure 4b). In the adjacent area of the stage 3 wound, fibers were stained fuchsia to red in Herovici’s staining, which was also consistent with IHC staining results showing obvious type I collagen signals (Figure 4c). Liver tissue with fibrotic lesions here was a positive control to address type III collagen specificity (Figure 2). In brief, the consistency between Herovici’s and IHC staining in this study suggests that Herovici’s staining on dolphin skin is applicable, relatively quick, and simple for distinguishing type I and III collagen. Furthermore, Herovici’s staining can be used in pathological diagnosis, e.g., kidney fibrosis or systemic multiple sclerosis.

In humans and mice, type III collagens show in blue fiber exist in the papillary dermis in Herovici’s staining [31]. In contrast, type III collagens were barely seen in the papillary dermis in the normal skin of Fraser’s dolphins; instead, type I collagens were in the majority. This difference might be relevant to different living niches and animal behaviors between terrestrial and marine mammals. Fraser’s dolphins are active and energetic swimmers, usually in tight, fast-moving schools, distributed in tropical, sub-tropical, and occasionally in warm-temperature waters [35]. Fraser’s dolphins can dive up to 600 m or more, which is indicated by its prey preferences of mesopelagic fish, cephalopods, crustaceans, etc. They tend to select larger prey inhabiting deeper water compared to other dolphins [35]. Therefore, a high abundance of stiff type I collagen in the normal skin of Fraser’s dolphin might provide higher tissue stiffness for maintaining mechanical properties of skin, e.g., skin tension, streamlining body, the resistance of water pressure or shearing force, and multiple external insults.

Type III collagen is involved in the early phase of wound healing [14]. In Fraser’s dolphin, type III collagens were observed in the granulation tissue in the stage 3 wound (Figure 3). Type III collagen has been found in scarless healing animal models with high levels, such as spiny mouse and human fetal skin [22,23,24,25,26], providing a softer wound bed for tissue remodeling. Though type III collagen could barely be seen in the normal skin of Fraser’s dolphins, it presented in wound edges of granulation tissues and papillary dermis under the neo-epidermis of the epithelial migrating tongue. This finding suggests the similar role of type III collagens in spiny mouse and human fetal skin, providing flexibility and a softer dermis environment for tissue reorganization. In stage 4 (immature healed) wound, thin and dense collagens were found in the wound center without adipocytes (Figure 3i and Figure 5a). Interestingly, thin collagen bundles with mixed type I and III collagens ran parallel to the surface epidermis, resembling human hypertrophic scar in the results of excessive collagen deposition after large full-thickness wound healing [36].

In Fraser’s dolphins, the hypertrophic-like wounds seem to be transient and would change to normal skin configuration over time. The parallel-organized collagens would be replaced by thicker collagen bundles, mainly composed of type I collagens and high amounts of adipocytes. The collagens’ excessive depositions in stage 4 wounds resulted from fibroblast production in the early wound-healing phase. Unlike the reported scarless wound-healing animals with a high level of type III collagen in normal skin and wound tissue, the high amount of type I collagen in Fraser’s dolphins might be responsible for maintaining skin integrity and mechanical support underwater pressure or shearing force. However, excessive collagen deposition with no adipocytes most likely influences the swimming performance with body flexibility and streamlining in fast swimming and thermoregulation and insulation when deep diving. Hence, a small amount of type III collagen might provide softness and flexibility to wound tissues to compromise the lack of adipocytes. Therefore, it is essential and rational for dolphins to deplete excessive collagens and eventually restore the skin configuration with appropriate collagen composition and adipocyte distribution. Despite that, this crucial step of how dolphins manipulate collagen degradation by matrix metalloproteinases (MMPs) and adipocyte restoration from stage 4 to stage 5 remains unknown. The removal of the hypertrophic-like wound in dolphin skin merits further investigation, which may be helpful for clinical use in human-injured skin repair.

Cetaceans are diverse in morphology, swimming behavior, and living environment. It has been reported that cetacean species exhibit fast and remarkable healing capacity [37], and they can live with numerous wounds, performing fast swimming or deep diving as usual. Some deep-diving cetaceans, for instance, sperm whales and beaked whales, can commonly dive exceed 1000 m deep. The skin stiffness and collagen compositions might be different from other shallow diving species, and so does collagen dynamics during wound healing. In addition, within one individual, collagen types and hypodermis blubber may exist differences in different body regions, e.g., back, belly, melon, or fins. The diversity in inter-/intra-cetacean species implies the different collagens’ structures and composition. Further research can work on other cetacean species or different body regions to have a better understanding of the role of collagens in the physical behavior and functional adaptation strategy to the aquatic environment.

## 5. Conclusions

In this study, the results demonstrate the changes in collagen during wound healing. The significant finding in Fraser’s dolphins’ skin included: (1) type I collagen was the main type of collagen; (2) type III collagen presented in early wound stages, and type I collagen increased later and eventually restored to the normal amount and configuration; (3) hypertrophic-like scar showed in the immature healed wound (stage 4) temporarily during wound healing. These results highlight that the great healing ability and functional restoration of the skin in Fraser’s dolphins may relate to collagen dynamics and appropriate elimination of excessive collagen deposition during wound tissue remodeling. Further studies to elucidate the mechanisms of collagen degradation for tissue remodeling after re-epithelization can provide a better understanding and improve the strategy for the clinical treatment of scars.

## Figures and Tables

**Figure 1 animals-13-01681-f001:**
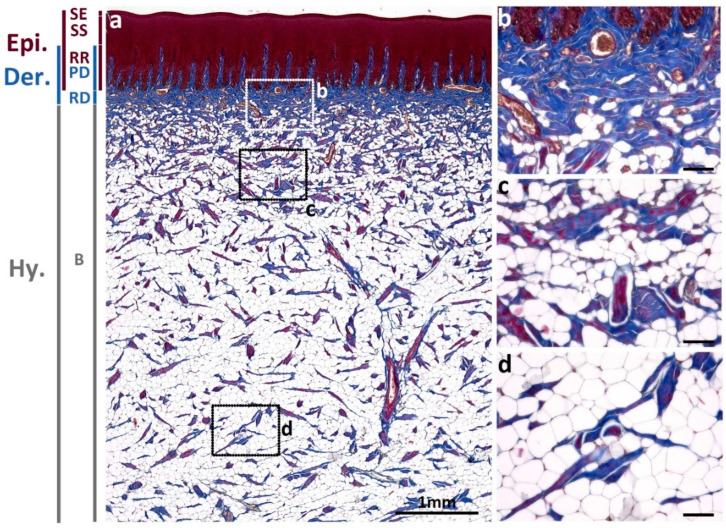
Overview of the epidermis, dermis, and blubber in normal skin of a sub-adult male Fraser’s dolphin (**a**). The blue color in trichrome staining represented collagen fibrils mainly located in the papillary dermis (PD) and reticular dermis (RD) and less in the blubber layer (B). The cytoplasm in the epidermis (Epi) is stained red, including stratum externum (SE), stratum spinosum (SS), and rete ridges (RR). Magnified figures showed (**b**) reticular dermis, (**c**) upper blubber layer, and (**d**) lower blubber layer. Dermis (Der). Hypodermis (Hy). Scale bars in (**a**) = 1 mm; in (**b**–**d**) = 100 μm.

**Figure 2 animals-13-01681-f002:**
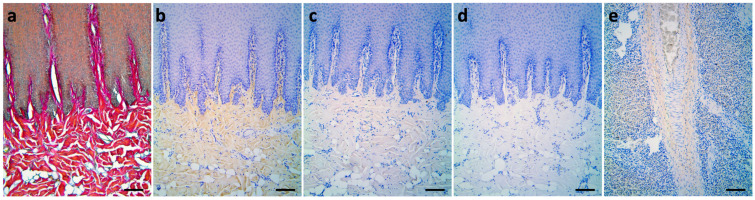
Differential staining of type I and type III collagens in normal skin of Fraser’s dolphins. (**a**) In Herovici’s staining, collagen fibers in the papillary dermis, reticular dermis, and blubber of normal skin were stained fuchsia to red, with no blue fiber observed. (**b**–**e**): Immunohistochemical staining. (**b**) Type I collagen signals (brown) existed in the papillary dermis, reticular dermis, and whole blubber layer. (**c**) Type III collagen signals were barely visible. (**d**) Negative control of type III collagen. (**e**) Liver tissue with fibrotic lesions served as a positive control for type III collagen. Scale bars = 100 μm.

**Figure 3 animals-13-01681-f003:**
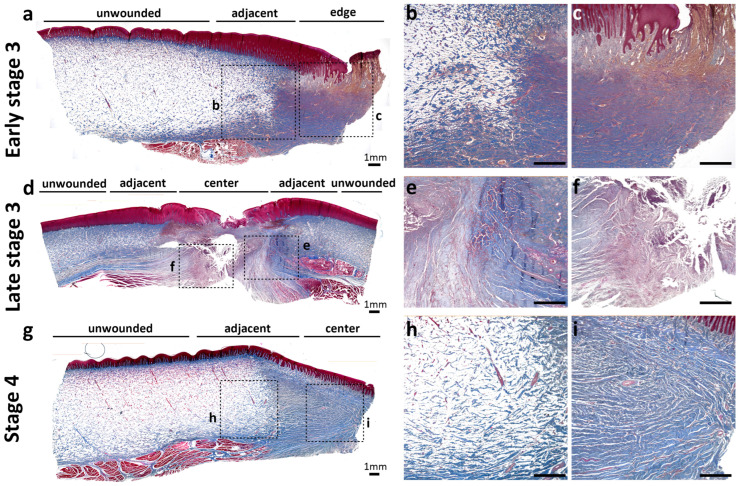
Collagen distribution during wound healing in Fraser’s dolphins. The definition of the wound healing stage is based on [21]. Open healing wound, early stage 3 (**a**–**c**); nearly closed wound, late stage 3 (**d**–**f**); immature healed wound, stage 4 (**g**–**i**) were stained with trichrome to compare collagen distribution. The blue color in trichrome staining indicates collagen fibril, and the red color represents cytoplasm in the epidermis and muscle tissue underlying blubber. Scale bars = 1 mm.

**Figure 4 animals-13-01681-f004:**
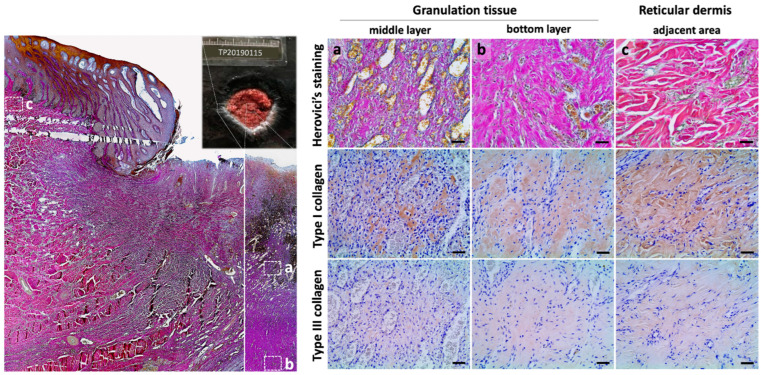
Differential staining of type I and type III collagen in stage 3 wound in Fraser’s dolphins. Left panel: low power view of stage 3 wound in Herovici’s staining. Particular regions: (**a**) The middle layer of granulation tissue; (**b**) the bottom layer of granulation tissue; (**c**) the reticular dermis at the wound adjacent are enlarged in the right panel. Right panel: differential staining of type I (red) and type III (blue) collagen with Herovici’s staining and IHC staining (brown). Scale bars = 50 μm.

**Figure 5 animals-13-01681-f005:**
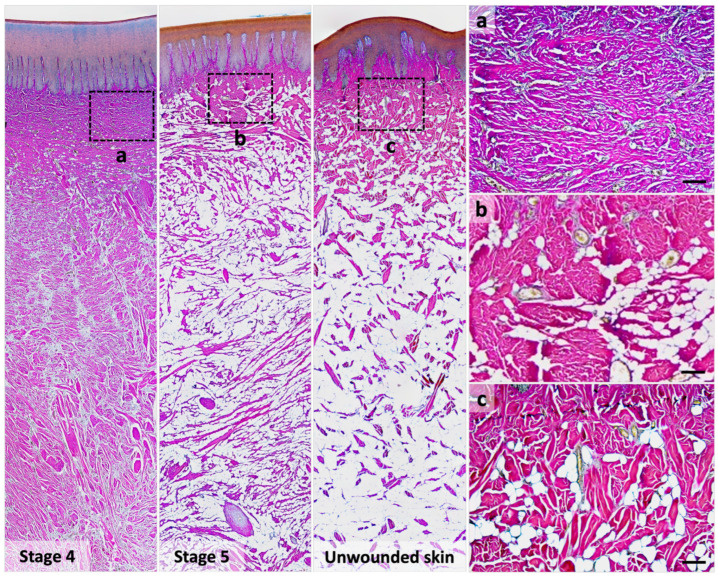
Differential staining of type I and type III collagens in the healed wound and unwounded skin in Fraser’s dolphins. In Herovici’s staining, the collagen fibers in the reticular dermis of the stage 4 wound (immature healed wound) were stained purple to fuchsia (**a**), whereas those in the stage 5 wound (mature healed wound) and the unwounded skin were stained fuchsia to red (**b**,**c**). Scale bars = 100 μm.

**Figure 6 animals-13-01681-f006:**
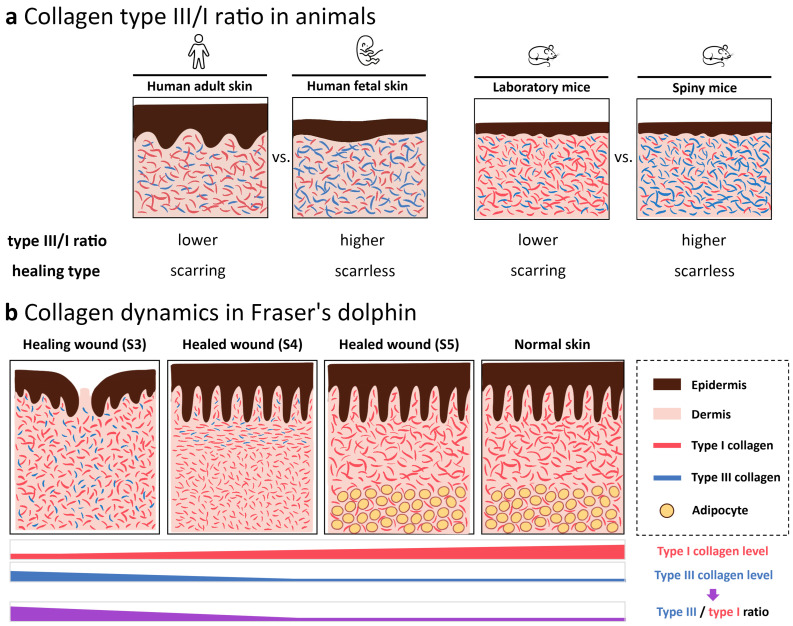
Illustrative diagrams of collagen type III/I ratio in animal skin. (**a**) Comparison of collagen type III/I ratio in scarring or scarless healing animals. In humans, the collagen type III/I ratio is higher in fetus skin than in adult skin, both in normal skin and during wound healing [25,26]. In mice, the collagen type III/I ratio in spiny mice in the early wound healing stages is higher than in the laboratory mouse [22,24]. (**b**) Changes in collagen composition during wound healing of stages 3 (healing wound), 4 (immature healed wound), and 5 (mature healed wound). Red fibril, type I collagen; blue fibril, type III collagen, yellow circle, adipocytes. These schematic diagrams, not drawn to scale, represented the epidermis and upper reticular dermis in different animals.

**Table 1 animals-13-01681-t001:** The details of each animal and the number of skin samples used in the current study.

Animal ID	Sex	Age	Body Length	Carcass Condition	Number of Samples
TD20181128	M	Adult	250 cm	Freshly dead	Stage 5 wound: 1
TP20190115	M	Adult	247 cm	Freshly dead	Stage 3 wound: 1 Stage 4 wound: 8 Stage 5 wound: 5
TT20190326	M	Sub-adult	200 cm	Freshly dead	Normal skin: 2 Stage 4 wound: 1
IL20191105	M	Sub-adult	221 cm	Freshly dead	Normal skin: 2 Stage 4 wound: 4 Stage 5 wound: 3
PT20201109	M	Sub-adult	189 cm	Freshly dead	Normal skin: 2 Stage 3 wound: 1
HL20201112	F	Calf	122 cm	Freshly dead	Stage 3 wound: 1

Note. The individuals TP20190115 and PT20201109 had a systemic infection.

**Table 2 animals-13-01681-t002:** Expression of type I and type III collagen in normal skin of Fraser’s dolphins.

Staining on Normal Skin	Type I Collagen	Type III Collagen
Herovici’s staining	+high amount (fuchsia to red)	−low amount (no blue)
IHC staining	+(in the whole dermis)	−(barely seen)

**Table 3 animals-13-01681-t003:** Type I and type III collagen expresses differentially in wounded skin tissue in Fraser’s dolphins. +, low coverage; ++, middle coverage; +++, high coverage.

Wounded Skin Tissue	Granulation Tissue		Reticular Dermis
Stage 3 Wound	Upper/Middle Layer	Bottom Layer	Adjacent Area
Herovici’s staining	purple + blue	purple/fuchsia	fuchsia to red
IHC staining type I	+	++	+++
IHC staining type III	+	+	+

**Table 4 animals-13-01681-t004:** Type I and III collagen dynamics during the ECM remodeling phase of the wound healing process.

Healed Wound	Immature Healed (Stage 4)	Mature Healed (Stage 5)	Unwounded Skin
Herovici’s staining	purple/fuchsia	fuchsia to red	fuchsia to red
Collagen type	type I+ type III	mainly type I	type I
Collagen bundle	thin	thin + thick (main)	thick

## Data Availability

Data is contained within the article.

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
