# Peer review of "Histopathological Study on Collagen in Full-Thickness Wound Healing in Fraser’s Dolphins (Lagenodelphis hosei)"

_animals, 2023, doi:10.3390/ani13101681_

Round 1
Reviewer 1 Report
This is a carefully done study and the findings are of considerable interest. A few minor revisions are list below:
1. Please briefly explain the reasons for choosing this species of dolphin and whether there is any uniqueness in the collagen changes of the wound.
2. In the keyword section, it is recommended to remove the "1. Introduction".
3. The author mentioned in the introduction that "Wound healing is... achieved through three... phases: inflammation, proliferation, and remodeling. "However," immature healed (stage 4) "and" mature healed (stage 5) "are used to describe the stage of wound healing in later texts. It is recommended to modify. Alternatively, address the necessity to use different terms.
4. In the introduction, when referring to the changes of collagen in human skin (from fetal to adult), please more clearly indicate the correlation between this change and this study.
5. "In contrast, type III collagens remain the main subtype in human fetal skin, which has scarless healing properties (line 125) "and"... type III/I collagen ratio...... play an important role in scarring/scarless fate determination. (line 126) ". These two are contradictory, so it is suggested to modify "scarless/scarring fate determination." or "type III/I collagen ratio".
6. It is mentioned that ”... help us gain more knowledge of... mechanical proper "(line 131), but no studies have been conducted. Numerical description of the changes in mechanical properties corresponding to the changes in collagen is suggested.
7. Whether the time from death to specimen collection of each stranded dolphin is approximately the same, and whether the specimen has the same freshness (or tissue activity).
8. The format of figure 2 is inconsistent with other figures. It is suggested to add a, b, c...
9. To what specific stage do "early stage 3" (line 233) and "late stage 3" (line 236) refer? It is suggested to describe it in regular terms.
10. Why is the collagen dynamic part (line 230) only involved in the early stage 3 and late stage 3? Whether slide staining at other stages of healing was performed.
11. It is also important to consider incomplete coverage during the healing stage (limited to S3-S5).
12. Please clarify whether the sample size is sufficient. Only one sample was taken at a single stage. It is important to consider whether there was any potential for individual bias in this situation. How to control bias.
There are no obvious problems with the grammar. But please have someone competent in the English language and the subject matter of your paper go over the professional vocabulary.
Author Response
Dear Reviewer,
We would like to take this opportunity to express our sincere thanks to you for identifying areas of our manuscript that needed corrections or modification. We uploaded the file of the revised manuscript that shows the highlighted text and a point-by-point response to your comments.
- Please briefly explain the reasons for choosing this species of dolphin and whether there is any uniqueness in the collagen changes of the wound.
Over the past few years, we have collected skin samples from several cetacean species. Among them, Fraser's dolphin provides us with the opportunity to conduct a comprehensive investigation into wound healing progression due to the relatively complete collection of samples, ranging from recently created to fully healed wounds. The uniqueness of the collagen changes observed in Fraser's dolphin has been described in the Discussion section (line 365-377). - In the keyword section, it is recommended to remove the "1. Introduction".
Thanks for your kind reminder! It has been deleted. - The author mentioned in the introduction that "Wound healing is... achieved through three... phases:
inflammation, proliferation, and remodeling. "However," immature healed (stage 4) "and" mature healed (stage 5) "are used to describe the stage of wound healing in later texts. It is recommended to modify. Alternatively, address the necessity to use different terms.
Thanks for your suggestion! This has now been amended (line 234-240), describing the definition of five stages in wound healing progress in Fraser’s dolphins. - In the introduction, when referring to the changes of collagen in human skin (from fetal to adult), please more clearly indicate the correlation between this change and this study.
Thanks for your suggestion! This has now been amended (line 113-129 & Figure 6a), describing the role of collagen type III/I ratio in scarring or scarless fate decision. - "In contrast, type III collagens remain the main subtype in human fetal skin, which has scarless healing properties (line 125) "and"... type III/I collagen ratio...... play an important role in scarring/scarless fate determination. (line 126) ". These two are contradictory, so it is suggested to modify "scarless/scarring fate determination." or "type III/I collagen ratio".
Thanks for your suggestion! This has now been amended (line 113-129 & Figure 6a), describing the role of collagen type III/I ratio in scarring or scarless fate decision. - It is mentioned that ”... help us gain more knowledge of... mechanical proper "(line 131), but no studies have been conducted. Numerical description of the changes in mechanical properties
corresponding to the changes in collagen is suggested.
Thanks for your suggestion! This has now been amended. (line 131-135) - Whether the time from death to specimen collection of each stranded dolphin is approximately the same, and whether the specimen has the same freshness (or tissue activity).
Thanks for your suggestion. Table 1 has now been amended. The information about carcass condition has been included in Table 1. - The format of figure 2 is inconsistent with other figures. It is suggested to add a, b, c...
Thanks for your suggestion! This has now been amended. (Figure 2) - To what specific stage do "early stage 3" (line 233) and "late stage 3" (line 236) refer? It is suggested to describe it in regular terms.
Thanks for your suggestion! This has now been amended (line 234-240), describing the definition of five stages in wound healing progress in Fraser’s dolphins. - Why is the collagen dynamic part (line 230) only involved in the early stage 3 and late stage 3?
Whether slide staining at other stages of healing was performed.
Stage 3 to 5 wounds were included in the current study. To avoid misunderstanding, the subtitle of 3.2.1 and 3.2.2 have been amended. (line 233, 265) - It is also important to consider incomplete coverage during the healing stage (limited to S3-S5).
The wounded skin analyzed in the current study did not include stages 1 (open wound) and 2 (initially healing wound without granulation tissue) because collagen deposition starting from stage 3. - Please clarify whether the sample size is sufficient. Only one sample was taken at a single stage. It is important to consider whether there was any potential for individual bias in this situation. How to control bias.
Thanks for your reminder! We have updated Table 1 to include the number of samples at each stage that were analyzed in the current study.
Reviewer 2 Report
The authors present a thorough histochemical and immunohistochemical analysis of dolphin skin with focus on type I and III collagens and their proportions during wound healing. Overall, this manuscript was easy to read and follow with descriptive methods and results. The figures and tables are well done and adequately labeled and described in the legends.
Some of the terminology and sentence structure in the abstract and introduction was a little awkward (e.g. Line 15 "leading to the unsuccessfully regain of skin junction").
One of the major questions was how the stages of healing determined in the dolphins utilized for the study? I understand some of these cases were reported in previous publication but perhaps a little more information in present publication would make it easier to compare across cases/stages. Was there evidence of underlying infection or colonization by opportunistic organisms that would alter the wound healing timeline? Was there inflammation? A more in depth description of the wounds for each case is warranted (e.g. routine HE stained sections with microscopic descriptions, also provide gross images in supplementary).
These results will be of value and interest to the readership, particularly for those who perform diagnostic histopathology on managed and free-ranging odontocete cetaceans with presumed similar cutaneous wound healing processes and capabilities.
After careful review of grammar and sentence structure, this manuscript will be ready for publication.
Author Response
Dear Reviewer,
We would like to take this opportunity to express our sincere thanks to you for identifying areas of our manuscript that needed corrections or modification. We have uploaded the file of the revised manuscript with highlighted text.
We have made amendments to the summary, abstract, and Table 1. Table 1 now includes information on carcass condition, health condition, and the number of skin samples used. We have also added a description of the five stages of wound healing progress in Fraser's dolphins in section 3.2.1 (line 234-240). Additionally, we have revised the description of the staining color of collagen fibers in Herovici's staining to include the range of colors observed, from fuchsia to red, instead of just red.
The wounded skin samples used in the current study were obtained from dead stranded dolphins. We directly compared the findings across stages and cases, and try to get more understanding on the dynamics of collagen during the wound healing process in dolphins. Unlike laboratory animals, however, continuous monitoring of the wound healing process is not feasible in our study. Due to this limitation, we would not be able to discuss the effect or fate of scarring when the animal has bacterial infection during early healing stage. It may be possible in the future when we have a live-stranded dolphin under treatment.